# Correlation between the Leaflet–Annulus Index and Echocardiographic Indices in Maltese Dogs with Myxomatous Mitral Valve Disease

**DOI:** 10.3390/vetsci10080493

**Published:** 2023-08-01

**Authors:** Han-Joon Lee, Hyung-Jin Park, Joong-Hyun Song, Kun-Ho Song

**Affiliations:** 1College of Veterinary Medicine, Chungnam National University, 99 Daehak-ro, Yuseong-gu, Daejeon 34134, Republic of Korea; hanjoon1102@cnu.ac.kr (H.-J.L.); jh.song@cnu.ac.kr (J.-H.S.); 2Sungsim Animal Medical Center, 131, Gyeryong-ro, Yuseong-gu, Daejeon 34134, Republic of Korea; gismo79@hanmail.net

**Keywords:** leaflet–annulus index, myxomatous mitral valve disease, echocardiography, Maltese, dogs

## Abstract

**Simple Summary:**

The present study evaluated the leaflet–annulus index depending on the stages of myxomatous mitral valve disease in Maltese dogs. As the disease worsened, the index significantly decreased. The correlation between the leaflet–annulus index and radiographic and echocardiographic indices, including vertebral heart score, vertebral left atrial size, left atrium to aortic root ratio, and left ventricular internal diameter at diastole normalized for body weight, was analyzed. They showed a negative correlation, with the left-atrium-to-aortic-root ratio demonstrating the strongest correlation with the leaflet–annulus index. Being one of the indicators and monitoring tools for mitral regurgitation in human medicine, the leaflet–annulus index can be adopted for veterinary medicine and could be a useful tool to determine the severity of mitral regurgitation and monitor its prognosis in Maltese dogs.

**Abstract:**

Myxomatous mitral valve disease (MMVD) is the most common chronic heart valve disease, leading to left-sided cardiomegaly in dogs. The leaflet–annulus index (LAI) was originally used in humans as a predictor of mitral regurgitation (MR) after mitral valve repair surgery. This index represents the quantity and severity of MR since it is affected by annular dilation. Recently, LAI was adapted to veterinary medicine, and its usefulness as an indicator of annular dilation on 2D transthoracic echocardiography in MMVD dogs was suggested. For this study, 135 Maltese dogs were selected and divided into groups of control, B1, and B2, according to the American College of Veterinary Internal Medicine consensus statement. The following data were collected: radiographic indices including the vertebral heart score and vertebral left atrial size, echocardiographic indices including the left-atrium-to-aortic-root ratio (LA:Ao), left ventricular internal diameter at diastole, normalized for body weight, and anteroposterior length and LAI measured on right parasternal long-axis view. The results showed a significant difference in LAI between each group, becoming smaller as the disease progressed. Also, there was a significant correlation between LAI and each index, showing the strongest correlation with LA:Ao. LAI could be helpful as a new indicator used for the determination of severity and prognosis in Maltese dogs with MMVD.

## 1. Introduction

Mitral regurgitation (MR) caused by myxomatous mitral valve disease (MMVD) is known as the most common acquired cardiac disease in dogs [1]. The pathophysiology of mitral regurgitation is due to the failure of the effective coaptation of anterior and posterior mitral leaflets [2]. The mitral valve competency is maintained by all components forming the mitral valve apparatus, such as both anterior and posterior mitral leaflets and papillary muscle [3]. If a myxomatous change occurs in the mitral leaflets, they become thickened and distorted [3,4]. These changes disrupt the organized function of the mitral valve apparatus, affecting the annular size, position of papillary muscles, and coaptation of valve leaflets [3]. Annular dilation is one of the major factors determining the severity of MR, and as myxomatous degeneration of mitral valves continues, regurgitation worsens [3,5]. Furthermore, the left atrial and ventricular remodeling exacerbate MMVD and cause changes on the thoracic radiography and echocardiography, eventually leading to cardiac dysfunction and heart failure with clinical signs [5]. 

In dogs, MMVD is diagnosed, classified into stages, and managed according to the American College of Veterinary Internal Medicine (ACVIM) consensus guideline statements [5]. The categorization of stages B1 and B2 is based on the criteria of murmur intensity, echocardiographic left atrial to the aortic root (LA:Ao) ratio, left ventricular internal diameter in diastole, normalized for body weight (LVIDDn), and radiographic vertebral heart score (VHS) [5]. Stage C and D are categorized according to the presence of the clinical signs of congestive heart failure [5]. Also, new indices in thoracic radiography, such as radiographic vertebral left atrial size (VLAS) or radiographic left atrial dimension (RLAD), are being researched in association with ACVIM criteria for more detailed staging of MMVD [1,6]. 

The leaflet–annulus index (LAI) was originally used in humans as a predictor of MR after mitral valve repair surgery [7]. LAI is the ratio calculated with the lengths of the anterior and posterior mitral leaflets (AML, PML) and anteroposterior length (APL) via two-dimensional (2D) transesophageal echocardiography (TEE) [7]. It represents the quantity and severity of MR since it is affected by annular dilation [2,7]. Recently, LAI was adapted to veterinary medicine and showed a significant correlation with the LVIDDn and LA:Ao ratio between ACVIM MMVD stages, suggesting its usefulness as an indicator of annular dilation being easily measured on 2D transthoracic echocardiography (TTE) [8]. 

However, the previous study was conducted with dogs of various breeds, which might have diverse index values depending on a body weight or heart size. Thus, our investigation focused solely on dogs belonging to the Maltese breed since it has a genetic predisposition to MMVD and is one of the most common breeds being diagnosed with MMVD in South Korea [9]. The purpose of this study is to analyze the difference in LAI compared to the ACVIM MMVD stage and find out the correlations between LAI and other common indices measured with thoracic radiography and 2D TTE in only a single Maltese breed.

## 2. Materials and Methods

### 2.1. Animals

The medical records of Maltese dogs that were admitted to the Chungnam National University Veterinary Medicine Teaching Hospital from 2017 to 2022 were collected and analyzed. The dogs who had been diagnosed with ACVIM MMVD stage B1 and B2 through physical examination, including cardiac auscultation, thoracic radiography, and 2D TTE, were selected as the subjects of this study. For the study, 135 Maltese dogs with MMVD were chosen. Stage B1 subjects were asymptomatic dogs with mitral regurgitation caused by MMVD [5]. Stage B2 subjects were dogs who met the criteria of ACVIM consensus guidelines (murmur intensity ≥ 3/6; echocardiographic LA:Ao ratio ≥ 1.6; LVIDDn ≥ 1.7; VHS ≥ 10.5) [5]. The dogs who underwent thoracic radiography and echocardiography but were not diagnosed with MMVD were also included as the normal control group. The exclusion criteria were concurrent congenital cardiac defects such as patent ductus arteriosus or ventricular septal defect that could cause left cardiac volume overload, prolapse, or flail movement of the mitral valve and other endocrinal diseases that can affect cardiac functions. There were 16 dogs in the control group, 40 dogs in B1 group, and 79 dogs in B2 group. The specific data of the dogs including sex, age, and body weight are presented in Table 1.

### 2.2. Thoracic Radiography

Thoracic radiographic images were obtained to measure VHS and VLAS on the right lateral view. A digital radiography system (MDXP-40TG, Medien International Co., Anyang-si, Republic of Korea) and viewing software (ZeTTa PACS Viewer, TY Soft, Anyang-si, Republic of Korea) were used for all measurements. For VHS, the length of the long axis of the heart from the ventral border of the bronchus to the most ventral margin of the cardiac apex and the length of the maximal short axis, perpendicular to the long axis, were measured [10]. Both lengths were positioned on the cranial edge of the 4th thoracic vertebral body and summed, which was calculated to be VHS (Figure 1) [10]. To measure VLAS, a line was drawn from the ventral aspect of the carina to the caudal aspect of the left atrium silhouette intersecting the dorsal margin of the caudal vena cava [6]. Then, the line was drawn from the cranial edge of the 4th thoracic vertebral body with the same length (Figure 2) [6]. All the measurements of VHS and VLAS were conducted by a single person.

### 2.3. Echocardiography

All dogs included in this study underwent 2D TTE (GE vivid E90, GE Healthcare, Seoul, Republic of Korea) examination. The examination was performed following recommendations from the Echocardiography Committee of the Specialty of Cardiology, ACVIM [11]. Mitral valve lesions were assessed with a 2D scan and a MR with color Doppler scan. The LA:Ao ratio was measured at the aortic root level in early diastole with a 2D scan, and LVIDD was measured with an M-mode scan at the papillary muscle level in end-diastole, both on the right parasternal short axis view [12,13]. Then, measured LVIDD values were normalized for body weight according to the formula: LVIDDn = LVIDD (cm)/[Body weight (kg)]^0.294^ [13]. Each index was measured five times, and the mean values of LA:Ao and LVIDDn from each examination were used. All the measurements of AML, PML, and APL were conducted by a single person via a 2D scan at the right parasternal long-axis view of LVOT in end-diastole. And, LAI was calculated according to the following formula: LAI = [AML (mm) + PML (mm)]/APL (mm) (Figure 3) [8].

### 2.4. Statistical Analysis

Statistical analysis was performed using a commercial software program (IBM SPSS Statistics 25.0, SPSS Inc., Chicago, IL, USA). Fisher’s exact test and a one-way ANOVA were performed for the homogeneity of age, body weight, and sex between groups. A Shapiro–Wilk test was performed for the normality test of indices. For data which were not normally distributed, they were presented as median (interquartile range), and when data were normally distributed, they were presented as mean ± SD. A Kruskal–Wallis test was performed to analyze the difference in the results from three groups, with Bonferroni correction for post hoc analysis. Spearman’s rank correlation coefficient was used to examine the relationship between variables and LAI. A *p*-value < 0.05 was considered statistically significant.

## 3. Results

The age, body weight, and sex showed no significant difference when compared between the control, B1, and B2 groups (Table 2).

Differences in the radiographic and echocardiographic indices between groups were analyzed (Table 3). The B2 group had the highest median of VHS compared to that of the control group and the B1 group with significant differences. The VHS difference between the control group and the B1 group showed no significance. Comparing VLAS measurements, the B2 group had the highest median of VLAS followed by the B1 group and the control group with a significant difference. The results of the LA:Ao ratio showed the highest values in the B2 group and lowest values in the control group, and there were significant differences between each group. The B2 group resulted in the highest median value in LVIDDn, being significantly higher than that of the control group and the B1 group. No significance was confirmed between the control group and the B1 group. The median APL of group B2 had the highest value compared to the control group and the B1 group with significance. The difference between the APL of the control group and the B1 group showed no statistically significant difference. The B2 group showed the lowest value of LAI, the highest value in the control group, and showed a significant difference between each other. The results are visualized, and graphs of each index are represented in Figure 4.

The correlation between LAI and other indices showed statistically significant negative correlations. The correlation coefficient values between LAI and the other indices are shown in Table 4. Among these correlations, the coefficient value between LAI and LA:Ao was the lowest compared to other indices. The graphs of correlation are represented in Figure 5.

## 4. Discussion

This study was conducted to analyze the difference in LAI between different stages of MMVD and correlation between LAI and other radiographic and echocardiographic indices. 

In human medicine, LAI has been used to predict the extent of MR in patients with mitral insufficiency, with LAI decreasing as the degree of MR increases [7,14]. Additionally, LAI is used as a prognostic indicator in mitral valve repair surgeries in humans [7]. As mitral valve repair is also being studied in veterinary medicine, establishing the differences in LAI based on MMVD stages and determining the criteria can provide a foundation for future research in this field [8]. The results of the LAI analysis revealed that there were differences in LAI according to the ACVIM stages in dogs, and similarly to humans, LAI decreased as the MMVD worsened [8]. However, the previous study had some limitations with regard to different breeds and a wide range of body weight in the dogs studied [8]. Therefore, we collected single Maltese dogs as the subjects for this study. As such, we created a control group and ACVIM stage B1 and B2 groups. Stage C–D was excluded from this study because it was difficult to apply LAI when it was highly likely to be accompanied by a saddle-shaped mitral valve or severe prolapse [2]. Since all the dogs belonged to the same breed, the groups were uniform when age and body weight were compared.

In this study, the VHS of the control group was significantly lower than that of the B1 group and the B2 group, and the B1 group showed a significantly lower VHS than the B2 group. This is because the patient group was formed according to the ACVIM consensus standard and divided according to the enlargement of the heart from stage B2, resulting in the observed differences [5,10].

The relationship between VLAS and cardiac enlargement differences according to the ACVIM stage has been demonstrated in other studies, and as the severity of MR increases, VLAS also increases [2,5,6]. According to the results of Kim’s study [15], the significance of the VLAS difference between the B1 and B2 groups was confirmed in Maltese dogs, but the significance between the control and the B1 groups was not recognized, as opposed to the significance recognized in this study. Based on the results of significant differences between each group in this study and Kim’s study, VLAS has the potential to be an efficient indicator of cardiac size for distinguishing ACVIM stage B1 and B2 specifically in Maltese dogs [15].

In this study, the LA:Ao was found to be lowest in the control and highest in B2 groups, with significant differences among all three groups. The post hoc analysis confirmed a significant increase in the LA:Ao ratio from the control group to the B1 group, and from the B1 group to the B2 group. Although study from Isaka et al. did not find a significant difference in LA:Ao values between the control and B1 groups, other studies have reported significant difference in LA:Ao values between control group and B1 group as up in this study [8,16]. Therefore, further research with a larger sample size is needed. 

In this study, LVIDDn in the B2 group was significantly higher than that of the control and B1 groups. No significant difference was observed between the control group and B1 group. Previous studies of Isaka et al. and Ogawa et al. also reported no significant difference in LVIDDn between a control group and B1 group in various breeds of dogs [8,17]. Since LVIDDn is a normalized value based on body weight, it can be applied to various breeds, and it is presumed that the significant difference observed between stage B1 and B2 is a consequence of the cardiac enlargement caused by mitral regurgitation.

Additionally, APL is the length measured by connecting the beginnings of the anterior and posterior mitral leaflets, and its value increases as the left atrium dilates due to regurgitation [14]. In this study, the APL showed the lowest value in the control group followed by the B1 and B2 groups with statistical significance. There was no significant difference between the control and B1 groups in terms of APL. As the stage of MMVD advanced, a previous study had shown that the APL became longer [8]. This is thought to be due to the widening of the annular ring that constitutes the mitral valve, and it is presumed to increase with the MMVD progression [14]. However, there was no significant difference between the healthy and B1 groups [8].

LAI was found to be related to APL, and LAI decreased as the severity of MR increased [8]. In this study, the results showed that the LAI of the control group was the highest, followed by the B1 group and B2 group in descending order, and the differences between the control and the B1 and B2 groups were significant. In a previous study, the LAI of the B2 group was significantly lower than that of the control and B1 group [8]. Additionally, no significant difference in LAI was observed between the healthy and B1 groups in the previous study [8], whereas this study showed a significant difference in the LAI between the control and B1 groups. This difference in findings may be attributed to differences in breed and sample size. 

Additionally, the correlations between LAI and thoracic radiographic and echocardiographic indices were confirmed in this study. In the correlation analysis of LAI, the results showed a negative correlation between LAI and each of the following: VHS, VLAS, LA:Ao, and LVIDDn. LA:Ao had the strongest negative correlation with significance. LAI is an indicator that can reflect left cardiac enlargement caused by MR [14]. As MR increases, the MMVD stage advances, and radiographic and echocardiographic indices, including VHS, VLAS, LA:Ao, and LVIDDn, also increase [5]. However, LAI decreases because the mitral annulus is distorted due to MMVD [14,18]. When MMVD progresses, impacting the left ventricular filling and left atrial pressure, the size of the left atrium expands, increasing the annular ring size [14,18]. This might explain the strong negative correlation between LAI and LA:Ao, as well the negative correlation between LAI and VHS, VLAS, and LVIDDn.

The limitation of this study is the small number of subjects. Although it was conducted on a single breed, further research with larger sample sizes is needed to address discrepancies of echocardiographic indices in previous studies and to include a diverse range of breeds in LAI research. Also, there is a lack of intra-observer variation analysis as well as a lack of information regarding the agreement between 3D-TTE- and TEE-derived LAI. In conclusion, the results of this study have shown the differences in LAI based on MMVD stages, suggesting that LAI can be used to predict the degree of MR associated with coaptation changes. These findings are expected to be helpful as a new indicator for determining the severity of MR and monitoring the prognostic assessment of MMVD in Maltese dogs. 

## Figures and Tables

**Figure 1 vetsci-10-00493-f001:**
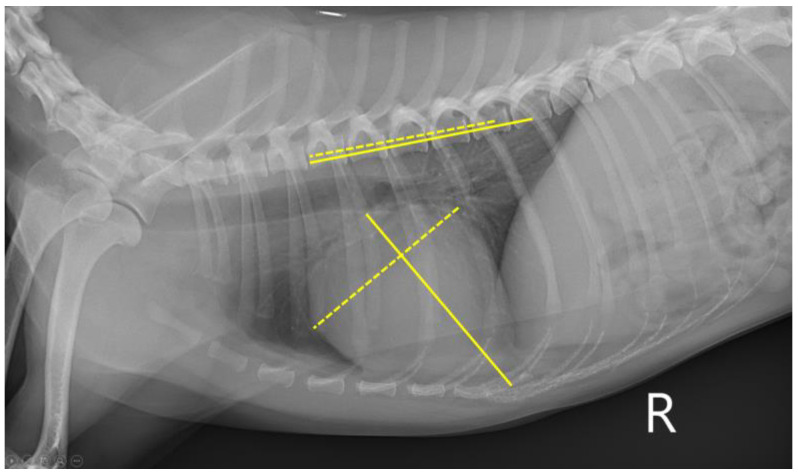
Radiographic measurement of VHS at right lateral thorax. VHS, vertebral heart score.

**Figure 2 vetsci-10-00493-f002:**
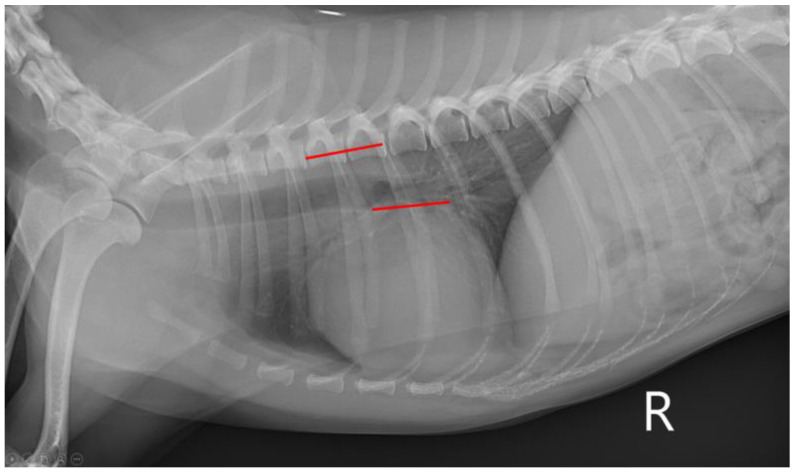
Radiographic measurement of VLAS at right lateral thorax. VLAS, vertebral left atrial size.

**Figure 3 vetsci-10-00493-f003:**
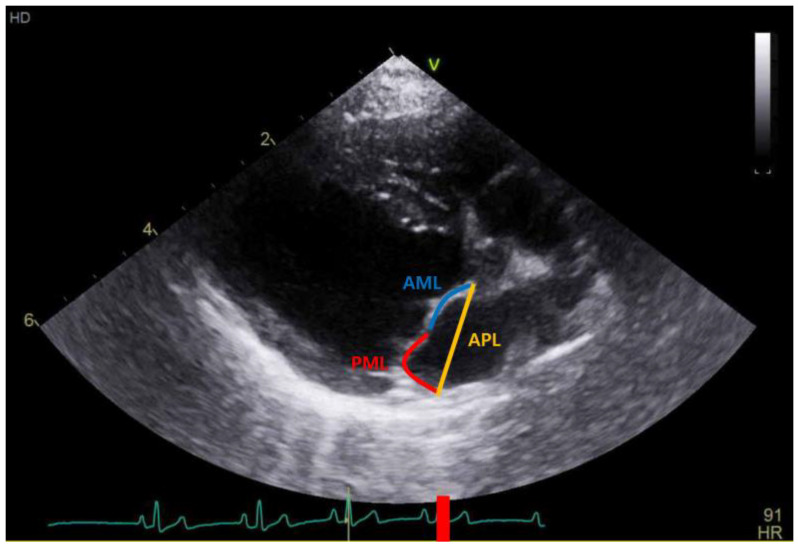
Leaflet–annulus index measurement on echocardiography at right parasternal long-axis view. AML, anterior mitral leaflet; PML, posterior mitral leaflet; APL, anteroposterior length.

**Figure 4 vetsci-10-00493-f004:**
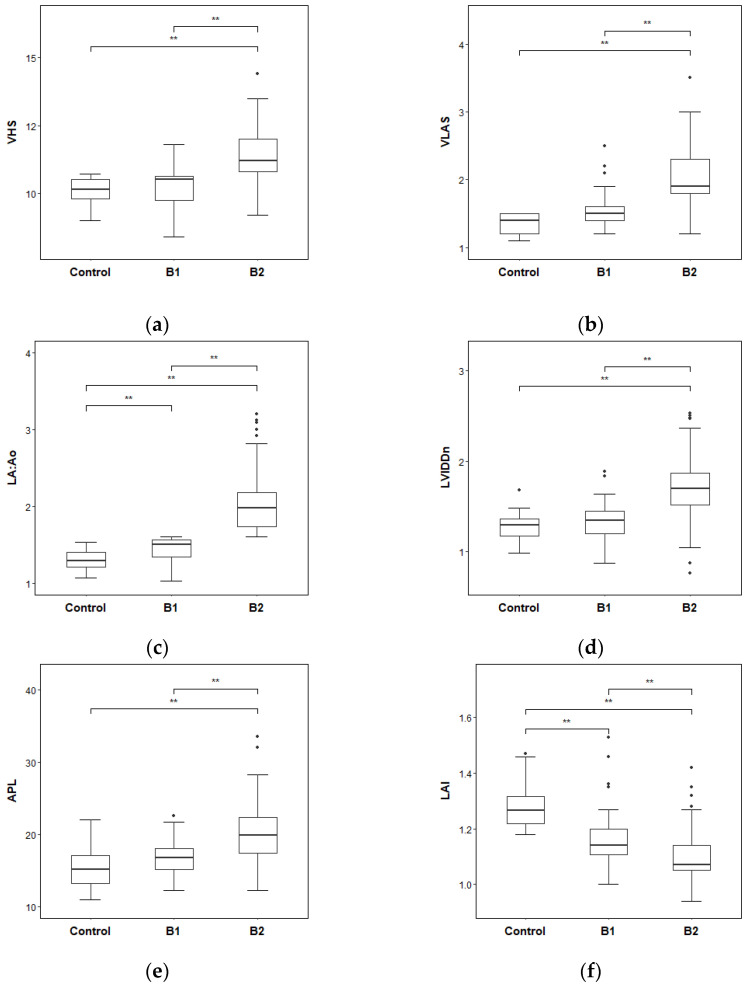
The differences in radiographic and echocardiographic indices between groups: (**a**) VHS; (**b**) VLAS; (**c**) LA:Ao; (**d**) LVIDDn; (**e**) APL; (**f**) LAI. ** *p* < 0.01. VHS, vertebral heart score; VLAS, vertebral left atrial size; LA:Ao, left-atrial-to-aortic-root ratio; LVIDDn, left ventricular internal diameter in diastole, normalized for body weight; APL, anteroposterior length; LAI, leaflet–annulus index.

**Figure 5 vetsci-10-00493-f005:**
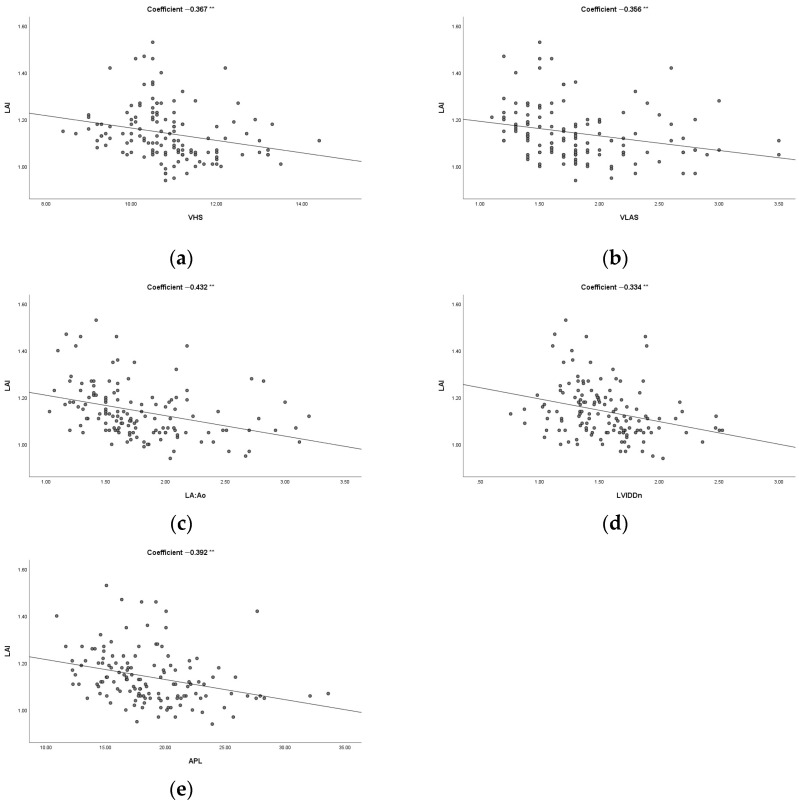
Graphs of correlation of LAI between radiographic and echocardiographic indices: (**a**) VHS; (**b**) VLAS; (**c**) LA:Ao; (**d**) LVIDDn; (**e**) APL. ** *p* < 0.01. VHS, vertebral heart score; VLAS, vertebral left atrial size; LA:Ao, left-atrial-to-aortic-root ratio; LVIDDn, left ventricular internal diameter in diastole, normalized for body weight; APL, anteroposterior length; LAI, leaflet–annulus index.

**Table 1 vetsci-10-00493-t001:** Specific data of the subjects.

Group (N) ^1^	Age (Year) ^2^	Body Weight (kg) ^2^	Sex (N) ^1^
Control (16)			Male (0)
	12.81 ± 2.48	3.52 ± 1.56	Female (2)
	(8.00–16.00)	(1.65–6.80)	Castrated male (10)
			Spayed female (4)
ACVIM B1 (40)			Male (0)
	12.13 ± 3.31	3.65 ± 1.09	Female (1)
	(5.00–20.00)	(1.90–6.65)	Castrated male (25)
			Spayed female (14)
ACVIM B2 (79)			Male (2)
	11.80 ± 2.78	3.54 ± 1.12	Female (7)
	(6.00–19.00)	(1.45–6.35)	Castrated male (37)
			Spayed female (33)
Total (135)			Male (2)
	12.01 ± 2.91	3.57 ± 1.16	Female (10)
	(5.00–20.00)	(1.45–6.80)	Castrated male (72)
			Spayed female (51)

^1^ Values represent the N, ^2^ Values represent the mean ± SD.

**Table 2 vetsci-10-00493-t002:** The differences in sex, age, and body weight between each group.

Characteristics		Total (N = 135)	Group			*p*
			Control (N = 16)	B1 (N = 40)	B2 (N = 79)	
Sex ^1^	Male	2 (1.5)	0 (0.0)	0 (0.0)	2 (2.5)	0.404
	Female	10 (7.4)	2 (12.5)	1 (2.5)	7 (8.9)	
	Castrated male	72 (53.3)	10 (62.5)	25 (62.5)	37 (46.8)	
	Spayed female	51 (37.8)	4 (25.0)	14 (35.0)	33 (41.8)	
Age (year) ^2^		12.01 ± 2.91	12.81 ± 2.48	12.13 ± 3.31	11.80 ± 2.78	0.431
Body weight (kg) ^2^		3.57 ± 1.16	3.52 ± 1.56	12.13 ± 3.31	3.54 ± 1.12	0.864

^1^ Values represent the N (%); ^2^ values represent the mean ± SD.

**Table 3 vetsci-10-00493-t003:** The difference in radiographic and echocardiographic indices between each group.

Index	Control ^a^ (N = 16)	B1 ^b^ (N = 40)	B2 ^c^ (N = 79)	*p*	Post Hoc
VHS	10.15 (9.60–10.50)	10.50 (9.58–10.68)	11.20 (10.80–12.00)	<0.001	a,b < c ^α,γ,ε^
VLAS	1.40 (1.20–1.50)	1.50 (1.40–1.60)	1.90 (1.80–2.30)	<0.001	a,b < c ^α,γ,ε^
LA:Ao	1.30 (1.20–1.40)	1.50 (1.33–1.57)	1.98 (1.73–2.18)	<0.001	a < b < c ^β,γ,ε^
LVIDDn	1.29 (1.17–1.37)	1.34 (1.19–1.45)	1.70 (1.51–1.87)	<0.001	a,b < c ^α,γ,ε^
APL (mm)	15.15 (13.09–17.55)	16.74 (15.08–18.23)	19.89 (17.45–22.54)	<0.001	a,b < c ^α,δ,ε^
LAI	1.27 (1.21–1.37)	1.14 (1.10–1.20)	1.07 (1.05–1.14)	<0.001	c < b < a ^β,γ,ε^

Values represented as the median (interquartile range). ^α^ *p* > 0.05 between a and b; ^β^ *p* < 0.001 between a and b; ^γ^ *p* < 0.001 between a and c; ^δ^ *p* < 0.01 between a and c; ^ε^ *p* < 0.001 between b and c. VHS, vertebral heart score; VLAS, vertebral left atrial size; LA:Ao, left-atrial-to--aortic-root ratio; LVIDDn, left ventricular internal diameter in diastole, normalized for body weight; APL, anteroposterior length; LAI, leaflet–annulus index. a: the control group; b: B1 group; c: B2 group.

**Table 4 vetsci-10-00493-t004:** The correlation coefficient values of LAI in relationship with other radiographic and echocardiographic indices.

Variables	VHS	VLAS	LA:Ao	LVIDDn	APL
LAI	−0.367 **	−0.356 **	−0.432 **	−0.334 **	−0.392 **

Values represent the Spearman’s rank correlation coefficients. ** *p* < 0.01. VHS, vertebral heart score; VLAS, vertebral left atrial size; LA:Ao, left-atrial-to-aortic-root ratio; LVIDDn, left ventricular internal diameter in diastole, normalized for body weight; APL, anteroposterior length; LAI, leaflet–annulus index.

## Data Availability

Data are contained within the article.

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
