# Peer review of "Correlation between the Leaflet–Annulus Index and Echocardiographic Indices in Maltese Dogs with Myxomatous Mitral Valve Disease"

_vetsci, 2023, doi:10.3390/vetsci10080493_

Round 1
Reviewer 1 Report
Correlation between the leaflet-annulus index and echocardiographic indices in Maltese dogs with myxomatous mitral valve disease
General comments:
This study investigated the difference in LAI values among different stages of MMVD and correlation between LAI and other radiographic and echocardiographic parameters in Maltese breed with MMVD.
From my understanding, LAI was introduced as a novel echocardiographic index to be used before the mitral valve repair surgery in human medicine to predict the postoperative outcome.
The results of this study demonstrated significant correlations between LAI and commonly used radiographic and echocardiographic parameters, which similar results has already been reported in dogs with MMVD. I would like the authors to clarify the significance of using this novel parameter, LAI, instead of already existing radiographic and echocardiographic parameters.
Additionally, in terms of novelty of the study, a single breed of dogs was selected as opposed to a mixture of different breeds, since according to the authors, body weight or heart size may result in diverse index values. If so, the study should have been better if the LAI values were also investigated in another larger size breed, so that a comparison can be made between the small and large breeds of dogs and study the influence of the body size. Or at least, make a reference to the previous study and compare the two studies in terms of differences in the body weight and LAI values.
Specific comments:
Line 13-15: The sentence is incomplete.
Line 20-21: Is this referring to dogs?
Line 21-22: What does it mean by “a predictor of mitral regurgitation? Does it refer to the prediction of the development, progression, or prognosis of MR? Please specify.
Line 40: Perhaps “failure” is better suited than “prevention”.
Line 41: What does it mean by “during systolic function”, please elaborate.
Line 62: As mentioned above already, what does it mean by “a predictor of MR”?
Table 1: Please specify the total number of dogs in each group.
Line 97: For the measurements of VHS and VLAS, it may be helpful to insert a figure illustrating the measurements.
Line 112-113: Please rephrase the sentence.
Line 126-127: Where it says “homogeneity between groups”, is it referring to the differences of sex, age and body weight?
Line 128-129: It should include that for data which were not normally distributed, they were presented as mean (range). It should also state that when the data were normally distributed, they were presented as mean ±SD.
Line 135-136: Is there a reason for omitting the results for differences in sex between groups?
Line 143-145: Where the significant difference observed between all the groups?
Line 145-147: What does it mean by “in order of significance”? Where the significant difference observed between every group?
Line 151-152: Should it mean “statistically significant difference” rather than “relationship”?
Line 152-154: Again, what does it mean by “in order with significant difference”? Are they in the order of significant or the actual values of the LAI?
Table 3: It helps to state the actual p-values of the post-hoc test within the text in the results. It should prevent many confusions caused in the results as mentioned above.
Figure 1: There is no need to show in the figures where there are no significant differences for (a), (d) and (e).
Line 173-174: Did you mean to say “This study was conducted to analyze the difference in LAI between different stages of MMVD and correlation between LAI and other radiographic and echocardiographic parameters”? If so, please rephrase it.
Line 179-180: Has there been any previous studies which have looked at the influence of body weight, body size and body conformation, for example in humans?
Line 195-196: Could you explain why significant difference in VLAS was detected between control and B1 groups, while the difference was not significant in the study by Kim et al.?
Line 204-205: Again, could you think of a reason for the differences in the results in LA:Ao between the two studies?
Line 207-210: You are simply re-stating the results. Please discuss the significance of the results.
Line 223-224: Would conducting a larger study solve the problem of 2D TEE and 3D TTE not being routinely used in veterinary medicine?
Line 247-148: Does this mean that accurate measurements of LAI can only be obtained using 2D TEE or 3D TTE? If so, what is the point of this study?
Line 249-250: If there are strong correlations between LAI and other parameters of radiograph and echocardiography, why not simply use these parameters. What is the importance of knowing the LAI? From this study, it is difficult to know the significance of measuring the LAI, instead of the already existing parameters.
Author Response
We really thank you for taking your time to give us such a thorough review and kind comments. Please see the attachment.

Reviewer 2 Report
1. This is an interesting and simple study, providing information on a relatively novel index in veterinary cardiology, on a single dog breed (Maltese).
2. Since this was a study based on previous examinations, the criteria for image quality selection should be mentioned. It is not stated which of the investigators made the measurements, or if any measurements were drawn from file data (rather than being measured again as part of the study design). Also, it would be ideal to have an intra-observer variation analysis, in order to verify that the AML, PML and APL measurements are reproducible.
3. Paragraph – lines 140 to 155 is the statistical analysis of the measurements between groups. There isn’t any mention of AML and PML results or statistical analysis, which would be interesting, and is part of the study design.
4. Table 3 should include the AML and PML measurements and results of the relevant statistical analysis (comparisons between groups).
5. In line 195 – 196 of the Discussion, there isn’t any explanation offered for the observed difference in VLAS when controls and B1 dogs were compared.
6. Line 199 – 202 of the Discussion also states that there is a statistically significant difference between the control group and B1 for the LA/Ao. However, the only reference offered that has reported such a difference is one from 1985. There ought to be more modern, bigger studies using control populations that have made such a comparison. As such, an explanation of why this discrepancy has been observed in this study is also warranted.
7. As a whole, this is an interesting study that merits publication, but not in its current form. Major changes need to be made, especially in the Discussion part.
Line 71 - 72: “We investigated on the single Maltese breed…” should be changed to “Our investigation focused solely on dogs belonging to the Maltese breed…”
Line 74: “… the difference of LAI depending on the …” should be changed to “ … the difference of LAI compared to …”
Line 75: “… find out the correlations of LAI among other …” should be changed to “… find out the correlations of LAI with other …”
Line 80: “… medical records of Maltese breed who visited…” should be changed to “… medical records of Maltese breed dogs that were admitted to the...”
Line 83: comma is missing after “2D TTE”
Line 112: “… was examined… “ should be changed to “… was used for echocardiography…”
Line 120 – 122: You should include three images of measurements of AML, PML and APL on normal, stage B1 and stage B2 dogs, to help the reader visualize the measurements.
Line 135 – 136: “… showed no significance with the control…” should be changed to “… showed no significance when compared between…”
Line 170 – 171: the term “visualized graphs” is a pleonasm and should be changed to simply “Graphs” or “Graphical representation”.
Line 176: the word “and” is not necessary.
Line 184 – 185: The last sentence is redundant as it has been mentioned in the results section. It could be replaced by “Since all the dogs belonged to the same breed, the groups were uniform when age and body weight were compared”.
Line 208 – 209: This sentence is redundant. It describes exactly the same results as the previous sentence, in a different manner.
Line 210 – 212: This is a very peculiar sentence that doesn’t offer any relevant information (and could be misleading at the same time).
Line 212 – 213: This is a sentence that has been repeated 3 times up to this point (in previous paragraphs). It doesn’t need to be at the end of each paragraph but could be included in the final paragraph or part of the study limitations paragraph.
Line 222 – 224: The lack of difference in APL dimensions between controls and B1 should not be attributed to inadequate evidence and further larger, single-breed or 3D studies should not be mentioned as solutions to a problem here. It makes perfect sense that APL is caused by left atrial and left ventricular enlargement and this is enough to explain the aforementioned finding.
Line 233 – 234: Again, the last sentence of this paragraph is repetitive and should be part of a larger sentence in the study limitations paragraph.
Line 244 – 246: This sentence should be changed to “ This might explain the strong negative correlation between LAI and LA:Ao, as well the negative correlation between LAI and VHS, VLAS and LVIDDn”.
Line 247 – 250: The limitation paragraph should be more extensive and should include the lack of intra-observer variation analysis as well as the lack of information regarding the agreement between TTE- and TEE-derived LAI. At the same time, the conclusion is rather broad and suggests the possible use of LAI as a prognostic factor, even though this hasn’t been analyzed in the study (for instance, there isn’t any Hazard Ratio analysis).
1Even though the reader is able to understand the thought process and results , it is nonetheless advised that a native English-speaking author or collaborator reviews the manuscript.
Author Response

(The authors gave the same response as above.)

Round 2
Reviewer 1 Report
General comments:
The revised manuscript shows much improvement. Please see some minor suggestions listed below.
Specific comments:
Line 18-19: I do not agree that LAI can be useful for the diagnosis of MMVD, but rather it may be helpful in the determination of severity.
Line 33-34: Same as the comment above.
Line 78-80: For the purpose of the study, you may want to emphasize that it is focused on one breed, or else there is no difference from the previous study by Isaka et al.
Line 90-91: You might want to describe the stage B2 criteria so that the readers can follow without referring to the guideline.
Line 96-97: Please specify what the data is representing, for example is it mean ±SD or SEM.
Line 133-134: You should specify how many measurements were taken to obtain the mean values.
Line 159-160: Did you mean, “The age, body weight, and sex showed no significant difference when compared between the control, stage B1 and stage B2 groups”?
Line 224-226: Please delete phrase “the analyze” as it is repetitive.
Line 253-255: Could you elaborate on this point in detail? You are simply stating the results without any discussion.
Line 264-266: Again, please discuss rather than simply stating the results.
Author Response
Thank you very much for your comment. We were able to complete this because of your effort and comments. Please see the attachment.
